# Natural Endotoxemia in Dogs—A Hidden Condition That Can Be Treated with a Potential Probiotic Containing *Bacillus subtilis, Bacillus licheniformis* and *Pediococcus acidilactici*: A Study Model

**DOI:** 10.3390/ani11051367

**Published:** 2021-05-11

**Authors:** Maria-Catalina Matei, Sanda Maria Andrei, Victoria Buza, Mihai Sorin Cernea, Daria Antonia Dumitras, Daniela Neagu, Horatiu Rafa, Cristian Paul Popovici, Andrei Radu Szakacs, Adrian Catinean, Eugen Stefanut, Laura Cristina Stefanut

**Affiliations:** 1Department of Animal Physiology, Faculty of Veterinary Medicine, University of Agricultural Sciences and Veterinary Medicine Cluj-Napoca, Calea Manastur No. 3-5, 400372 Cluj-Napoca, Romania; catalina.matei@usamvcluj.ro (M.-C.M.); victoria.buza@usamvcluj.ro (V.B.); cristina.stefanut@usamvcluj.ro (L.C.S.); 2Department of Biochemistry, Faculty of Veterinary Medicine, University of Agricultural Sciences and Veterinary Medicine Cluj-Napoca, Calea Manastur No. 3-5, 400372 Cluj-Napoca, Romania; sandrei@usamvcluj.ro (S.M.A.); antonia.dumitras@usamvcluj.ro (D.A.D.); horatiu.rafa@usamvcluj.ro (H.R.); 3Department of Pharmacology, Faculty of Veterinary Medicine, University of Agricultural Sciences andVeterinary Medicine Cluj-Napoca, Calea Manastur No. 3-5, 400372 Cluj-Napoca, Romania; 4Department of Internal Medicine, Faculty of Veterinary Medicine, University of Agricultural Sciences and Veterinary Medicine Cluj-Napoca, Calea Manastur No. 3-5, 400372 Cluj-Napoca, Romania; daniela.neagu@usamvcluj.ro (D.N.); popovicivet@gmail.com (C.P.P.); 5Department of Animal Nutrition, Faculty of Veterinary Medicine, University of Agricultural Sciences and Veterinary Medicine Cluj-Napoca, Calea Manastur No. 3-5, 400372 Cluj-Napoca, Romania; andrei.szakacs@usamvcluj.ro; 6Department of Internal Medicine, Iuliu Hatieganu University of Medicine and Pharmacy, 400006 Cluj-Napoca, Romania; catinean@gmail.com; 7EST Research, 040251 Bucharest, Romania; eugen_stefanut@yahoo.com

**Keywords:** Bacillus spores, leaky gut in dogs, apparent dysbiosis, endotoxins

## Abstract

**Simple Summary:**

Digestive problems are a frequently occurring condition in dogs. Their manifestations can be clinically visible or, in some cases, hidden at the level of the digestive tract. Such a condition is represented by naturally occurring endotoxemia that may, or may not, have a clinical manifestation. However, as it is a little-studied condition in dogs, there are not enough data to diagnose it and its manifestation may be neglected in clinical practice. Probiotics are believed to be an alternative method that can be included in the treatment scheme for gastrointestinal (GI) problems. By their mechanism of action, those formulas may be able to improve or stop the symptoms of different digestive problems and also target the source of the problem. Probiotics can be considered to be host-friendly treatments, with beneficial effects for general health status. As probiotics do not represent aggressive treatments and have few or no side effects, using this type of alternative treatment can increase the animals’ welfare. The aim of the present study was to characterize naturally occurring endotoxemia in dogs and assess the effect of a probiotic formula on this condition. We suggest that this hidden condition can be treated with probiotics as an alternative that is friendlier for animals.

**Abstract:**

Spore-based *Bacillus* spp. products are considered to have a higher probiotic potential compared to products containing only lactic acid bacteria because their viability in the gastrointestinal (GI) tract is higher, even when GI environmental conditions are unfavorable. The aim of this study was to assess the effect of a *Bacillus subtilis*, *Bacillus licheniformis* and *Pediococcus acidilactici* spore-based potential probiotic on the natural levels of postprandial endotoxemia. A total of 11 dogs completed the study: group 1—healthy dogs: *n* = 5; group 2—dogs with apparent dysbiosis: *n* = 6. For 30 days, the dogs were fed the probiotic product; clinical examinations and blood sampling were done before and after completion of the probiotic treatment. Endotoxin levels were assessed pre-meal, 6 h and 12 h post-meal, before initiation and after completion of the treatment. The results showed a decrease in endotoxin levels after treatment, especially 12 h post-meal (group 1: 20.60%; group 2: 44.93%). This study reports new information with regard to natural endotoxemia levels in dogs and suggests that a multi-strain formula (spore-based) consisting of *B. subtilis*, *B. licheniformis* and *P. acidilactici* is able to diminish endotoxin values.

## 1. Introduction

Probiotics formulations contain live microorganisms or components of microbial cells that are able to provide a beneficial effect in the host when administered in adequate amounts. The mechanisms by which these beneficial effects are achieved include reduced intestinal permeability, increased mucin secretion by goblet cells, increased production of defensins that prevent the colonization of pathogens, increased short-chain fatty acid production, stimulation of IgA secretion, decreased luminal pH and increased tolerance of immune cells for commensal microorganisms while maintaining the ability to respond to pathogens [1,2]. Nonviable microorganisms are also considered capable of conferring beneficial effects by adhering to the mucus layer of the gastrointestinal (GI) tract and stimulating immune functions [2].

Probiotics are able to produce beneficial effects in the host without permanently modifying the microbiome, which is likely because of transient colonization in the intestine [2,3].

Members of the Bacillus genus are considered to have a higher probiotic potential compared to lactic acid-producing bacteria. This is explained by the fact that they have a high level of viability and are able to reproduce and increase in number in the GI tract even in unfavorable environmental or low pH conditions. Whether this feature provides an advantage for these microorganisms in relation to their potential probiotic effects is controversial, as many researchers argue that a bacterium does not have to be viable to exert probiotic effects [4,5]. However, the pathogens of Bacillus sp. have been understood to be opportunist pathogens over the last decade [6]. It is important to take into consideration that probiotic safety requirements are strain-dependent [7,8]. Accordingly, not all the bacteria from Bacillus sp. have probiotic characteristics. *Pediococcus acidilactici* is also a probiotic bacterium with beneficial effects on the health of the gastro-intestinal tract [9]. *P. acidilactici* belongs to a lactic acid-producing bacterium category that has a wide range of benefits in dogs [10]. However, probiotic characteristics are strain-dependent and, therefore, studies on the same species cannot be translated directly [7,8].

Currently, probiotics are considered to be a potential alternative treatment for dogs with gut problems. Ailments such as leaky gut or intestinal dysbiosis may be treated using such formulas. Moreover, Bacillus spores are thought to have the ability to diminish endotoxemia levels. Endotoxemia is a condition that is present in all mammals. It is characterized by an increase in serum endotoxin levels. It can affect either GI permeability, the GI microbiota or both [11]. In human medicine, it is known that this increase occurs about 5 h after ingestion of a meal and that it affects approximately 33% of the human population [12]. In the case of dogs, mild endotoxemia levels were reported after intravenous infusion of low-dose endotoxin [13] but naturally occurring endotoxemia has not been characterized. Experimentally induced endotoxemia has been studied in humans using either a bolus injection of *Escherichia coli* endotoxin [14] or by consuming a high-fat meal [11].

The aim of the present study was to identify the pattern of natural post-meal endotoxemia levels in clinically healthy dogs and those with apparent dysbiosis. Additionally, we aimed to test whether a spore-based potential probiotic formulation containing *B. subtilis*, *B. licheniformis* and *P. acidilactici* was able to reduce natural endotoxin levels after 30 days of treatment.

## 2. Materials and Methods

### 2.1. Study Design

This study was conducted at the University of Agricultural Sciences and Veterinary Medicine of Cluj-Napoca (Romania). The study protocol was designed according to the recommendations for pilot study criteria [15,16,17,18] and the rule of the three R’s suggested by ethics regulations (Replacement, Reduction and Refinement) [17]. As a result, the number of animals included in the study was selected to obtain sufficient data to answer the research question [19,20]. The investigations were performed in the departments of Animal Physiology, Internal Medicine, Biochemistry and Parasitology.

A total of 11 dogs were enrolled in the study. This study used real-world clinical cases (Figure 1). Throughout the course of the study, the dogs did not experience any lifestyle or nutritional changes. The two groups were formed based on specific inclusion/exclusion criteria characteristic for each group. Inclusion criteria for group 1 (healthy dogs) were that the dog exhibited no GI manifestations (diarrhea, vomiting), had no history of antibiotic treatment in the last 6 months, was clinically healthy and normally consumed one meal/day. Exclusion criteria for this group were the presence of GI manifestations (diarrhea, vomiting), endoparasites or a history of antibiotic treatment in the last six months. Group 2 (dogs with intestinal dysbiosis) was composed of dogs with GI manifestations (diarrhea, vomiting) or dogs that were under current antibiotic treatment (minimum duration of two weeks) and which were consuming one meal/day. Exclusion criteria for this group were intestinal endoparasites, acute live failure and acute kidney failure.

The dogs were given a potential probiotic product composed of three strains of bacteria: *B. subtilis* HU58, *B. licheniformis* SL-307 and *P. acidilactici* (FidoSpore^®^ supplied by Microbiome Labs, LLC, 101 E Town Place, Suite 210 Saint Augustine, FL 32092, USA). The product was given in capsules. One capsule was administered per day with the daily meal for 30 days. Liver powder was used to improve the taste of the probiotic product and to ensure it was ingested.

On day 0 of the study, dogs were clinically examined by a veterinarian and both fecal (in order to identify the presence of endoparasites) and blood samples were collected. For the detection of endotoxemia, blood samples were collected pre- and post-meal (6 h and 12 h). Between days 1 and 30, the probiotic product was administered to the animals according to the manufacturer recommendations. On day 31, all the dogs were clinically examined and fecal and blood samples were once again collected. For the detection of endotoxemia, blood was collected pre- and post-meal (6 h and 12 h) (Table 2).

### 2.2. Clinical Investigations

To form the study groups based on the inclusion and exclusion criteria, evaluate the health of the dogs entering the study and obtain baseline clinical data, each dog was given a clinical evaluation. A general clinical examination, which included collection of data related to clinical parameters (cardiac frequency, respiratory frequency, body temperature), was performed on the dogs. The normal range values for dogs in the Merck Veterinary Manual (2016) were considered normal clinical values in this study. 

### 2.3. Determination of Endotoxemia

The ToxinSensorTM Chromogenic LAL Endotoxin Assay Kit (GenScript, Piscataway, NJ, USA) was used to assay endotoxemia. This method utilizes a modified Limulus amebocyte lysate and a synthetic color-producing substrate to detect endotoxin chromogenically. The end product could be measured using a spectophotometer (545 nm). Endotoxin levels were determined using a standard curve. The minimum endotoxin detection limit for the kit was 0.01 EU/mL while the measurable concentration range was 0.01 to 1 EU/mL. 

Endotoxemia evaluation was performed for all dogs (group 1 and group 2), pre-meal and post-meal (6 h and 12 h), on day 0 (prior to probiotic administration) and day 31 (after completion of 30 days of daily probiotic administration). Endotoxemia was assessed using serum diluted as follows: undiluted, 1:1, 1:2.

### 2.4. Statistical Analysis

The obtained data were analyzed from a statistical point of view, even though the number of subjects (*n* = 11) was small. A conditional analysis of variance (ANOVA) was performed with the following conditions: healthy dogs or dogs with dysbiosis × experiment time (day 0 and day 31) × meal time (pre-meal, 6 h post-meal and 12 h post-meal). Significance was set at *p* < 0.05 (determined using the Student–Newman–Keuls multiple comparisons test). Changes in study parameters were determined using linear regression by trend line and expression of R^2^.

## 3. Results

### 3.1. Clinical Examination

A general clinical examination was performed on all the studied dogs before starting the treatment (day 0) and at the end of the study (day 31).

All dogs presented no significant treatment needs during the clinical examination. All study dogs had a normal temperature, ranging between 37.9 and 39.9 °C, and cardiac rates did not indicate any issues of significant clinical importance. All the studied dogs presented higher than normal values for respiratory rate (33–46 breaths per minute). The increased respiratory rate was not alarming and could be explained by excitement, stress and/or a high temperature in the examination room. 

The clinical appearance of group 1 (healthy dogs) was not altered by the treatment. No symptoms, such as diarrhea or vomiting, were observed in this group. 

For group 2 (dogs with dysbiosis), clinical examination revealed diarrhea, vomiting and/or skin lesions; the presence of GI symptoms qualified each dog for inclusion in the study. After completion of 30 days of the treatment, a decrease in digestive symptoms was observed and, in some cases, the disappearance of digestive problems. Improvements were observed in the general conditions of these dogs; the details of the improvements depended on the original diagnosis of each dog. 

### 3.2. Endotoxemia

Group 1 (healthy dogs): Before administration of the product, an increase in the mean endotoxin concentration at 6 h post-meal compared to the pre-meal mean concentration was observed (pre-meal, 0.465 ± 0.113 EU/mL; 6 h post-meal, 0.473 ± 0.172 UE/mL). The same dynamic was observed after completion of 30 days of product administration (pre-meal, 0.3677 ± 0.2266 EU/mL; 6 h post-meal, 0.3878 ± 0.2327 UE/mL) (Table 3 and Figure 2). These differences were not statistically significant because of the high level of variation between individual dogs. The average percent decreases in endotoxin concentrations from day 1 to day 31 are shown in Figure 2 and were as follows: pre-meal, 20.97%; 6 h post-meal, 15.00%; 12 h post-meal, 20.60%.

Group 2 (dogs with dysbiosis): The concentrations of endotoxemia pre- and post-probiotic administration indicated a decrease in the level of endotoxemia after completion of treatment (Table 3 and Figure 2). This implied a substantial improvement in intestinal conditions. The most important change observed was the decrease in endotoxin concentrations after the completion of 30 days of product administration, as measured 12 h post-meal (44.93%) (Figure 3). 

## 4. Discussion

Endotoxemia is a condition that affects the normal function of the gut. In veterinary medicine, metabolic endotoxemia has not been well studied [12]. Questions regarding endotoxemia, such as when it appears and how it affects canine digestion, are still outstanding. Currently available data are either extrapolated from human medicine or obtained from studies using dogs as a canine model for product testing. From such studies, it can be learned that a small amount of endotoxin, 0.1 g/kg, was able to produce mild endotoxemia in dogs [21]. Another study aimed to establish a canine endotoxemia model by injecting a single bolus of 0.03, 0.1 or 1.0 μg/kg body weight lipopolysaccharide intravenously, which resulted in mild endotoxemia [13]. However, the evolution of this condition may be different in naturally occurring cases compared to experimental ones. 

In humans, it is known that, approximately 5 h after meal consumption, an increase in endotoxin concentrations occur [12]. Findings from human studies related to post-meal endotoxemia cannot be extrapolated to dogs because the digestion process (i.e., the enzymes used and time needed to complete food digestion) varies between individuals and breeds. Moreover, the differences between the diets of dogs and humans are quite large, given that dogs are carnivores and humans are omnivores. 

The type of diet a dog is fed has a major influence on the level of natural endotoxemia that they experience. In the present study, patient 1.1 (group 1) followed a diet based on raw food, while the other four dogs in the group were fed with dry food. As an abrupt change in diet can produce GI imbalances, such as diarrhea or even vomiting, we did not change the diet of the dogs in our study. Dogs enrolled in the study followed their usual diet, with the same type of food and the same meal intervals as they were accustomed to. A close look at the endotoxemia results showed that there was a difference between patient 1.1’s endotoxemia levels and those of the other dogs in the healthy dog group. However, all five patients in this group met the inclusion criteria. The fact that patient 1.1 followed a diet based on raw food may have influenced the levels of endotoxin absorbed in the blood stream after consuming a meal.

Endotoxemia can be present in dogs without any clinical signs. Moreover, it can occur in clinically healthy individuals after consuming a meal and regardless of the fat composition of the meal. It was observed that the mean endotoxemia levels in group 1 increased 6 h post-meal vs. pre-meal levels. This observation was also valid for this group both before initiating the probiotic and after completion of 30 days of treatment. Although an increase was present in the mean endotoxin concentrations 6 h post-meal (vs. pre-meal) in the dogs after they completed the treatment, a significant decrease in the concentrations of endotoxin detected before and after completion of treatment was also observed. This decrease was observed at the pre-meal and post-meal (6 h and 12 h) time points. Comparing endotoxin concentrations at day 0 and day 31 showed that pre-meal concentrations decreased by 20.97%, concentrations at 6 h post-meal decreased by 15.00% and concentrations at 12 h post-meal decreased by 20.60%. These results support the initial hypothesis that this treatment is able to improve intestinal digestion in healthy individuals and, as a result, decrease inflammation and endotoxemia levels. 

The same pattern was observed in group 2. The levels of endotoxemia were higher at all three time points (pre-meal, 6 h post-meal and 12 h post-meal) before initiating the treatment compared with those obtained after completion of the treatment. Moreover, the average percent decreases were greater compared to those in the healthy group. We found a 25.89% decrease in pre-meal endotoxin concentrations (day 0 to day 31), a 35.00% decrease at 6 h post-meal and a 44.93% decrease at 12 h post-meal. This trend in decreased endotoxin concentrations after the treatment demonstrates the efficacy and impact that the treatment had on reestablishing homeostasis in the GI tract. 

Currently, there are no data available regarding postprandial natural endotoxemia levels in dogs. The main hypothesis of our study was extrapolated from human medicine, where such data are available [12]. However, the digestion process of dogs and humans is not identical. The amount of time in which partially digested food stays in the stomach of dogs is between 4 h and 8 h, varying somewhat between individuals and breeds, while for humans the digestion process lasts about 1 h. Another difference in the digestion process between the two species is the length of the GI tract in relation to the whole-body dimensions of the individual [22]. Such aspects may influence the time in which serum endotoxin is detectable. Considering such information, we decided to assess serum endotoxin levels at 6 h and 12 h post-meal consumption. In both groups of dogs, a rise in serum endotoxin concentrations was observed for all post-meal assessments (vs. pre-meal), with small variations. This finding supports the hypothesis that, after consuming a meal, serum endotoxin concentrations rise; however, these data do not establish the most appropriate timing of post-meal serum endotoxin assessment in dogs. 

The dynamic testing of serum endotoxin concentrations was conducted with the purpose of identifying the optimal time to detect post-meal endotoxin concentrations. After analyzing our results, we determined that there is no particular pattern or time point that is optimal for detecting post-meal endotoxin concentrations. Although we did not observe a peak in endotoxin increases at a single given time point, we did observe an increase in post-meal endotoxin concentrations in all the studied dogs. This observation supports the hypothesis that endotoxin concentrations rise after consuming a meal, which has already been demonstrated in humans [11,12]. Also, in order to achieve more impactful data, further research on the link between naturally occurring endotoxemia in dogs and other safe and relevant microbes with possible probiotic effects should be undertaken. 

## 5. Conclusions

The data obtained from this pilot study provide new information about natural endotoxemia levels in both healthy dogs and those with symptoms of GI dysbiosis. Moreover, based on the test results, it can be concluded that the multi-strain, spore-based potential probiotic test product was able to decrease post-treatment endotoxin concentrations.

## Figures and Tables

**Figure 1 animals-11-01367-f001:**
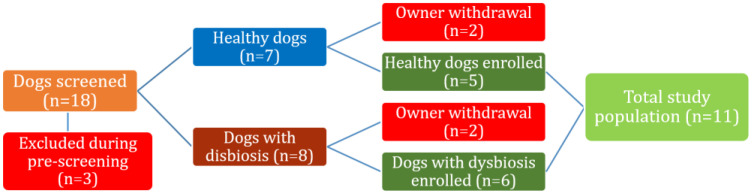
Patient disposition. Patients were carefully screened using the study exclusion/inclusion criteria; qualified dogs were enrolled in the study. Of the 18 dogs that were enrolled in the study, a total of 11 completed the entire study period: five healthy dogs and six dogs with apparent dysbiosis (Table 1).

**Figure 2 animals-11-01367-f002:**
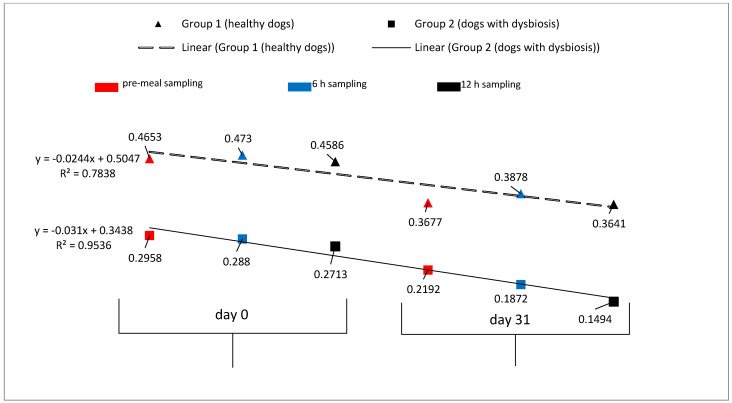
Changes in the mean concentrations of endotoxemia and their linear regression (EU/mL) in healthy dogs and dogs with dysbiosis (pre- and post-probiotic treatment).

**Figure 3 animals-11-01367-f003:**
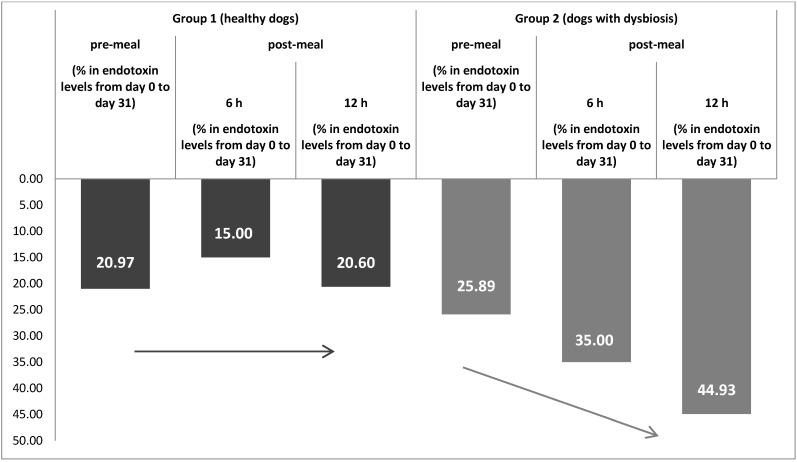
The percent decrease in endotoxin concentrations (EU/mL) from day 0 to day 31 in healthy dogs and dogs with dysbiosis (pre- and post-probiotic treatment).

**Table 1 animals-11-01367-t001:** Study participants.

Group 1			Group 2		
Patient Code ^1^	Age	Sex	Patient Code ^2^	Age	Sex
1.1	48 months	Male	2.1	7 months	Male
1.2	20 months	Female	2.2	69 months	Female
1.3	50 months	Female	2.3	4 months	Male
1.4	24 months	Female	2.4	24 months	Male
1.5	17 months	Female	2.5	53 months	Male
			2.6	36 months	Male

^1^ Group 1—healthy dogs; ^2^ group 2—dogs with intestinal dysbiosis.

**Table 2 animals-11-01367-t002:** Study schedule.

DAY/EXPERIMENTAL TIME		ACTIVITIES
Day 0	Blood collection 1: pre-meal	Clinical examination, collection of biological samples
	Blood collection 2: 6 h post-meal	
	Blood collection 3: 12 h post-meal	
DAY 1–DAY 30 → PROBIOTIC ADMINISTRATION		
Day 31	Blood collection 1: pre-meal	Clinical examination, collection of biological samples
	Blood collection 2: 6 h post-meal	
	Blood collection 3: 12 h post-meal	

**Table 3 animals-11-01367-t003:** Change in endotoxemia levels (EU/mL).

Endotoxemia Evaluation Moment		Group 1 ^1^ (*n* = 5)		Group 2 ^2^ (*n* = 6)	
		Pre-A ^3^ (Day 0)	Post-A ^4^ (Day 31)	Pre-A (Day 0)	Post-A (Day 31)
Pre-meal ^5^ mean ± SD		0.4653 ± 0.113	0.3677 ± 0.226	0.2958 ± 0.168	0.2192 ± 0.241
Post-meal ^6^ mean ± SD	6 h	0.4730 ± 0.172	0.3878 ± 0.232	0.2880 ± 0.200	0.1872 ± 0.145
	12 h	0.4586 ± 0.149	0.3641 ± 0.242	0.2713 ± 0.170	0.1494 ± 0.070

^1^ Group 1—healthy dogs; ^2^ group 2—dogs with intestinal dysbiosis; ^3^ measurement before the treatment (day 0 of the study); ^4^ measurement after the treatment (day 31 of the study); ^5^ before meal sampling; ^6^ after meal sampling. Mean—mean value; SD—standard deviation; *n*—number of studied dogs.

## Data Availability

The data presented in this study are available on request from the corresponding author. The data are not publicly available due to General Data Protection Regulation.

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
