# Peer review of "Natural Endotoxemia in Dogs—A Hidden Condition That Can Be Treated with a Potential Probiotic Containing Bacillus subtilis, Bacillus licheniformis and Pediococcus acidilactici: A Study Model"

_animals, 2021, doi:10.3390/ani11051367_

Round 1
Reviewer 1 Report
I think this pilot study is interesting because the topic is current and deals with a topic that has been studied in recent years. The use of probiotics is much discussed and this publication consider an innovative aspect, the anti-toxemia action. The organization ofe the study is correct and the results and discussion are adequate. The tables are clear. I have only one curiosity: the levels of endo toxemia are higher in group 1 /healty dogs) than in group 2 ( dogs with presumed dysbiosis), the Authors think that this is linked to different diets?
Author Response
The diet of the dogs may have an influence on the endotoxemia concentrations. Because an abrupt change in diet can produce GI imbalances such as diarrhea or even vomiting, we did not change the diet of the dogs in our study. In group 1 (healthy dogs) one of the subjects followed a raw based diet. That fact may have influenced the concentrations of endotoxin absorbed in the bloodstream. In our opinion, it could be a link between the concentrations of endotoxin and the different diets of the dogs.

Reviewer 2 Report
As this manuscript describes the use of three novel probiotic candidates for canine use, it is most important to recognise the nature of these bacteria species. Bacillus sp, including B. subtilis and B. licheniformis have been known to be oppostunist pathogens during the last decade (Logan, 1988) and new studies emphasize the pathogenic mechanisms of these species (Calendroni et al 2016). Probiotic safety requirements are strain dependent (Hill et al 2014, WHO 2002, EFSA 2021) and should be discussed also in this article to avoid misleading the reader to believe Bacillus sp have probiotic characteristics in all aspects, such as wording in R 69-R72. As the strains tested in this study don’t fulfill the probiotic definition, the word probiotic can’t be used to describe them.
The sporeforming characteristics is in the interest of industrial manufactures, not a quality needed for benefiting the animal health. All probiotics survive the GIT, as a requirement.
Also, note that probiotic characteristics is strain dependent. Therefore, studies with bacteria of same species even in dogs will not translate directly. Even less, if the studies have been conducted in other animal species, such as chickens (R75-R76).
With only 11 dogs in the study, there is very little to work with statistically. For this reason, please state this very clearly.
The study design and relevance is excellent, try reconstructing the study with safe and relevant microbes to achieve an impactful data (R276-R283). Mean while, this data needs to be addressed a study model more than a pilot study as with no safety data, there is no possibility to carry on with high risk strains.
Please, check the spelling errors.
Author Response
Point 1: As this manuscript describes the use of three novel probiotic candidates for canine use, it is most important to recognise the nature of these bacteria species. Bacillus sp, including B. subtilis and B. licheniformis have been known to be oppostunist pathogens during the last decade (Logan, 1988) and new studies emphasize the pathogenic mechanisms of these species (Calendroni et al 2016). Probiotic safety requirements are strain dependent (Hill et al 2014, WHO 2002, EFSA 2021) and should be discussed also in this article to avoid misleading the reader to believe Bacillus sp have probiotic characteristics in all aspects, such as wording in R 69-R72. As the strains tested in this study don’t fulfill the probiotic definition, the word probiotic can’t be used to describe them. The sporeforming characteristics is in the interest of industrial manufactures, not a quality needed for benefiting the animal health. All probiotics survive the GIT, as a requirement.
Response 1: We agree with the fact that Bacillus sp. can be opportunist pathogens and with the fact that probiotic safety requirements are strain dependent. It was added in the text an explanatory phrase that clears this aspect (R 75-R78). Regarding the use of the word probiotic, it was changed with “potential probiotic” in the whole manuscript, as long as we found your observation correct.
Point 2: Also, note that probiotic characteristics is strain dependent. Therefore, studies with bacteria of same species even in dogs will not translate directly. Even less, if the studies have been conducted in other animal species, such as chickens (R75-R76).
Response 2: This is a correct argument. It was added in the text an explanatory phrase in order to make this clear for the readers (R81- R82).
Point 3: With only 11 dogs in the study, there is very little to work with statistically. For this reason, please state this very clearly.
Response 3: The study population was small (n=11). We agree that from a statistical point of view, there are not many possibilities to analyze the data. According to your requirement, this explanation was also added in the text (R163- R164). [We would like to add that the decision to work on such a small number of subjects was made taken into consideration animals’ welfare and all the ethics regulations that suggest limiting as much as possible the number of subjects used in a study]
Point 4: The study design and relevance is excellent, try reconstructing the study with safe and relevant microbes to achieve an impactful data (R276-R283). Mean while, this data needs to be addressed a study model more than a pilot study as with no safety data, there is no possibility to carry on with high risk strains.
Response 4: Thank you for the appreciation of our work. We will take into consideration your suggestion and we will continue our research in the future, in order to achieve impactful data. We stated this also in the text (R287- R289). Meanwhile, for the data presented in this study, we changed the term “pilot study” to “study model” as you suggested.
Point 5: Please, check the spelling errors.
Response 5: The spelling errors were corrected in the whole text.
